# Glutathione Depletion and MicroRNA Dysregulation in Multiple System Atrophy: A Review

**DOI:** 10.3390/ijms232315076

**Published:** 2022-12-01

**Authors:** Chisato Kinoshita, Noriko Kubota, Koji Aoyama

**Affiliations:** 1Department of Pharmacology, Teikyo University School of Medicine, 2-11-1 Kaga, Itabashi, Tokyo 173-8605, Japan; 2Teikyo University Support Center for Women Physicians and Researchers, 2-11-1 Kaga, Itabashi, Tokyo 173-8605, Japan

**Keywords:** multiple system atrophy, neurodegenerative disease, glutathione, microRNA, oxidative stress, α-synuclein

## Abstract

Multiple system atrophy (MSA) is a rare neurodegenerative disease characterized by parkinsonism, cerebellar impairment, and autonomic failure. Although the causes of MSA onset and progression remain uncertain, its pathogenesis may involve oxidative stress via the generation of excess reactive oxygen species and/or destruction of the antioxidant system. One of the most powerful antioxidants is glutathione, which plays essential roles as an antioxidant enzyme cofactor, cysteine-storage molecule, major redox buffer, and neuromodulator, in addition to being a key antioxidant in the central nervous system. Glutathione levels are known to be reduced in neurodegenerative diseases. In addition, genes regulating redox states have been shown to be post-transcriptionally modified by microRNA (miRNA), one of the most important types of non-coding RNA. miRNAs have been reported to be dysregulated in several diseases, including MSA. In this review, we focused on the relation between glutathione deficiency, miRNA dysregulation and oxidative stress and their close relation with MSA pathology.

## 1. Introduction

Multiple system atrophy (MSA) is a rare, adult-onset, fatal neurodegenerative disease (ND) characterized by progressive loss of neuronal and oligodendroglial cells in various sites in the brain [1]. Disease onset is usually around 50 years old or later, with the age distribution of onset peaking in the late 50s. Currently, MSA is diagnosed when parkinsonism or cerebellar ataxia presents with prominent autonomic failure, and it is often accompanied by rapid eye movement (REM) sleep behavior disorder [2]. MSA belongs to the group of α-synucleinopathies, which are morphologically characterized by abnormal accumulation of fibrillary α-synuclein (α-syn) in the neurons and oligodendrocytes [3]. Oligodendroglial inclusions are specifically detected in the brains of MSA patients; these inclusions are the main hallmark of the disease and are considered to play a critical role in the primary events leading to MSA [4]. Although the origin of α-syn inclusions in the oligodendrocytes has been controversial, the leading hypotheses are that either internalization of neuron-secreted α-syn or de novo α-syn abnormality in oligodendrocytes is a key event in the pathogenic cascade leading to the propagation and spread of α-synucleinopathies of MSA [5]. Symptomatic therapies are available for MSA-associated parkinsonism and autonomic failures, but the response to these treatments is often poor [6]. Moreover, there is currently no effective medicine to ameliorate or even slow the progress of MSA. The rapid and devastating disease progression of MSA symptoms results in a relatively short survival period after disease onset [7]. Indeed, the estimated survival is approximately 9 years from symptom onset [8].

Because of the low incidence and uncertain diagnosis, there have been far fewer studies on MSA than on major NDs such as Alzheimer’s disease (AD), Parkinson’s disease (PD) and amyotrophic lateral sclerosis (ALS). Recently, however, improvements in the diagnostic certainty and advances in genetic technologies have promoted investigation of the etiological mechanisms of MSA. Although no hereditable mutations have been identified in MSA, some family history of genetic mutations as well as polymorphisms in unrelated patients with MSA have been found [9,10,11,12,13,14]. Recently, microRNA (miRNA) has emerged as a novel component of the gene expression regulation for maintaining brain functions [15]. Abnormalities in miRNA expression have been reported to cause several NDs, including MSA [16,17]. The first genome-wide miRNA analysis of MSA brain tissue revealed dysregulation of miRNAs, resulting in downregulation of the solute carrier family transporters for glutathione (GSH) and taurine, both of which are important antioxidants for neuroprotection against oxidative stress [18]. It is generally thought that the initiation and progression of NDs are likely induced by oxidative stress via an imbalance of oxidants and antioxidants [19,20]. GSH is known as a key protector against oxidative stress, and both nuclear magnetic resonance spectral and postmortem analyses have shown GSH depletion in the brains of patients with NDs [21,22,23,24,25,26,27,28]. In this review, we focus on the relation between miRNA dysregulation and oxidative stress, both of which appear to play a role in MSA pathology.

## 2. An Overview of MSA

MSA is a rare, rapidly progressing, fatal neurodegenerative disorder of uncertain etiology that is clinically characterized by a variable combination of parkinsonism, cerebellar impairment, autonomic failure and motor dysfunctions [29]. The term “multiple system atrophy” was coined by Graham and Oppenheimer in 1969, who proposed it as the combination of various separate neurological disorders into a single disease with symptoms and signs of lesions affecting several central nervous system (CNS) structures and multiple physiological systems [30]. Differential diagnosis between MSA and other forms of neurodegenerative parkinsonism such as PD is often difficult in practice. Useful biomarkers for clinically distinguishing MSA from other NDs are urgently needed.

### 2.1. Clinical Classification of MSA

Until fairly recently, MSA cases were classified into three types—olivopontocerebellar atrophy (OPCA), striatonigral degeneration (SND), and Shy-Drager syndrome (SDS)—which are characterized by prominent cerebellar and extrapyramidal signs, atypical parkinsonism, and parkinsonism with autonomic failure, respectively [31]. In 1998, consensus criteria for the clinical diagnosis of MSA yielded a simpler classification into two subtypes depending on whether the initial symptoms were predominantly parkinsonian (MSA-P) or cerebellar (MSA-C) [29,32,33] (Figure 1). MSA-P is associated with SND, the symptoms of which are similar to those of PD—namely, muscular rigidity, bradykinesia and postural instability—although resting tremor appears to be less severe in MSA-P than in PD. On the other hand, MSA-C is roughly synonymous with OPCA from the previous classification, the main symptoms of which are cerebellar ataxia characterized by difficulty coordinating walking, hand movements, speech, and eye movements. SDS is not included as a subtype in this new consensus criteria, but it is known to be a clinical form of MSA with autonomic failure as the primary symptom.

### 2.2. Pathological Features of MSA 

Pathologically, MSA is characterized by abnormal glial cytoplasmic inclusions (GCIs) and neuronal inclusions (NIs) of aggregated α-syn in several areas of the nervous system, including the cerebellar cortex, pons, striatum, substantia nigra and brain stem [34] (Figure 1). Most of the studies assessing the pathogenesis of MSA have focused on the mechanisms underlying intracellular accumulation of α-syn, an approximately 14.5-kDa protein composed of 140 amino acids that is physiologically expressed in the human brain [35]. Accumulating evidence has suggested that either α-syn is transferred from neurons to oligodendroglia or that α-syn pathology spreads in a prion-like manner [34]. Although the cause of misfolded α-syn deposition in the oligodendrocytes and neurons of patients with MSA is still obscure, it is known that the uptake, accumulation, and oligomerization of extracellular α-syn is promoted by oxidative stress in oligodendrocytes [36]. 

Oxidative stress is basically characterized by an imbalance of oxidants and antioxidants that results in an excess of reactive oxygen species (ROS) [37]. Because mitochondria are a major source and an immediate target of ROS, a vicious circle of oxidative stress and mitochondrial dysfunction may trigger and exacerbate NDs including MSA [38]. The majority of cellular ROS are generated by the mitochondrial respiratory chain and oxidative phosphorylation system, which consists of five complexes, I–V, through which electrons pass [39]. In Complex I, electrons borne on nicotinamide adenine dinucleotide dehydrogenase (NADH) are transferred to coenzyme Q10; in Complex II, electrons from succinate in the tricarboxylic acid cycle pass to coenzyme Q10. Complex III then passes the electrons on to cytochrome c, and Complex IV completes the chain, passing the electrons to O_2_, thereby reducing it to H_2_O. Finally, ATP is generated by the influx of these protons back into the mitochondrial matrix through ATP synthase (Complex V). The majority of physiological ROS production is generated from this electron transport system. Mitochondrial dysfunction possibly caused by genetic mutation and/or environmental stimuli provokes oxidative stress, and thus mitochondrial dysfunction may be associated with MSA pathogenesis [40].

MSA is classified into two subtypes depending on whether the initial symptoms are predominantly parkinsonian (MSA-P) or cerebellar (MSA-C). MSA is pathologically characterized by glial cytoplasmic inclusions (GCIs) and neuronal inclusions (NIs) containing abnormal aggregations of α-syn in several areas of the nervous system. In the brains of patients with MSA, α-syn aggregation in oligodendrocytes may cause myelin degeneration.

### 2.3. Distinction between MSA and Lewy Body Diseases

Since MSA (especially MSA-P subtype) shares many symptoms and signs with PD (e.g., parkinsonism, autonomic failure and REM sleep behavior disorder), it is often difficult to distinguish between them at an early stage even with recent diagnostic criteria [41,42]. Clinically, levodopa responsiveness is a supporting feature of PD while MSA is often unresponsive to levodopa [43,44]. MSA, PD, and other Lewy body diseases (LBD) form a group of synucleopathies characterized by pathologic aggregates of α-syn [45], mainly differing in the sites of α-syn aggregation in the brain. Glial cytoplasmic inclusions of α-syn are hallmarks of MSA while aggregates of α-syn in the perikarya and neurites of neurons are known as Lewy bodies and Lewy neurites, respectively, and are present in PD, PD with dementia and dementia with Lewy body [45,46]. Recently, accumulating evidence suggests the existence of distinct strains of α-syn, possibly because of genetic polymorphism or protein modification, and their association with different patterns of disease propagation and atrophic regions [47]. Interestingly, several studies have indicated that α-syn derived from GCIs of MSA has more potent activity than that from Lewy bodies of PD [47,48]. The diversity of seeding propensities of α-syn in different brain regions supports the notion of the clinical and morphological heterogeneity of MSA as well as clinical difference between MSA and PD/LBD [41,49]. In addition, different forms of α-syn aggregates may also affect the distinct pathogenesis and/or pathology between MSA and PD/LBD [50]. Interestingly, several studies have reported that an α-syn seeding assay using CSF, abdominal skin or olfactory mucosa of patients was able to detect α-synucleinopathies (see reference review [51] for details), which could be a valuable diagnostic tool in the future pending further studies and analysis.

## 3. Genes Associated with MSA Features

MSA (previously termed OPCA, SND and SDS) was primarily categorized into spinocerebellar degenerations (SCDs), which comprise a group of sporadic and hereditary NDs with lesions involving the cerebellum and spinal cord [51,52]. Among SCD subtypes, MSA has been recognized as a sporadic disease, and many of the dominantly inherited SCDs have been renamed spinocerebellar ataxia (SCA) [53]. Thus, MSA has not generally been considered a genetic disease. However, a few familial cases of MSA have recently been reported, and genomic analysis of these cases revealed that MSA is transmitted in an autosomal dominant or recessive inheritance pattern in some pedigrees (Table 1). The gene encoding α-syn (SNCA) is one of the candidate genes for causing MSA. A study of a British family with an SNCA variant revealed the neuropathological hallmarks of both MSA and PD with autosomal dominant parkinsonism [54]. In addition, a Finnish family with another SNCA variant showed neuropathological findings comparable with both PD and MSA, although the possibility that the two diseases may coexist cannot be excluded [14]. 

Unexpectedly, several studies identified recessive mutations of the gene encoding 4-hydroxybenzoate polyprenyltransferase (COQ2) in several unrelated Japanese families with MSA [9,55]. In addition, functionally impaired COQ2 variants were shown to be associated with sporadic MSA in a Japanese population [9,55]. The COQ2 gene encodes an enzyme essential for the biosynthesis of coenzyme Q10 [56]. Coenzyme Q10, in turn, plays a crucial role in mitochondrial electron transport; it transfers electrons from complex I and II to complex III as described in Section 2.2, and deficiencies in the transfer cause mitochondrial dysfunction and oxidative stress [57]. Moreover, it has been reported that cerebellar and plasma levels of coenzyme Q10 were lower, and the mitochondrial dysfunction and oxidative stress levels higher, in MSA cases compared to controls, indicating that COQ2 and MSA are strongly associated [58,59]. Although several reports attempting to identify a COQ2 mutation in patients with MSA have failed in this effort [11,60,61], mitochondrial dysfunction and/or oxidative stress in the brains of patients with MSA could play a role in MSA etiology. In addition, a recent study reported that, in rare cases, COQ2 variants are related to the onset of familial PD, suggesting that COQ2 variants might share similar pathways and be involved in induction of the phenotype of PD or parkinsonism [62].

Genes associated with other NDs, such as microtubule associated protein tau gene (MAPT), leucine-rich repeat kinase 2 (LRRK2), β-glucocerebrosidase (GBA1), TATA-box binding protein gene (TBP), coiled-coil-helix-coiled-coil-helix domain containing 2 (CHCHD2) and chromosome 9 open reading frame 72 (C9orf72), might be involved in the MSA pathogenesis [63,64,65,66,67,68], but more research is required to determine whether these genes have specific association with MSA pathology.

**Table 1 ijms-23-15076-t001:** List of genes that are mutated and/or altered in MSA.

Gene Name	Encoding Protein	Physiological Function	Pathology	Related NDs [69]
SNCA	α-synuclein	regulation of synaptic vesicles and neurotransmitter release	inclusion formation	PD, LBD
COQ2	polyprenyl transferase	catalyzing coenzyme Q10 biosynthesis	coenzyme Q10 deficiency, followed by mitochondrial dysfunction	PD
MAPT	microtubule-associated protein tau	formation and stabilization of axonal microtubules	tau accumulation	AD, FTD, PD
LRRK2	leucine rich-repeat kinase 2	involved in neuronal plasticity, autophagy, and vesicle trafficking	associated with pathologies of α-syn	PD
GBA1	β-glucocerebrosidase	hydrolysis of glucosylceramide and glucosylsphingosine	α-syn accumulation	AD, FTD, PD
TBP	TATA-box binding protein	component of the eukaryotic transcription initiation machinery	formation of aggregates	PD, HD
CHCHD2	coiled-coil-helix-coiled-coil-helix domain containing 2	regulating electron flow in the mitochondrial electron transport chain	mitochondrial dysfunction	AD, FTD, PD, ALS,
c9orf72	chromosome 9 open reading frame 72	regulation of autophagy and vesicular trafficking	formation of aggregates	FTD, ALS
ASCT1	alanine/serine/cysteine/threonine transporter 1	uptake of neutral amino acids	enhanced oxidative stress	No data
EAAC1	excitatory amino acid carrier 1	cysteine uptake	GSH depletion, followed by oxidative stress	No data
TauT	taurine transporter	taurine uptake	taurine depletion, followed by oxidative stress	No data
NOVA1	neuro-oncological ventral antigen-1	alternative splicing, involved in the formation and activity of the synapses	autoantigen in paraneoplastic opsoclonus myoclonus ataxia	No data
Oct1	organic cation transporter1	Translocation of organic cations across the blood–brain barrier	unknown	No data

Abbreviations: AD; Alzheimer’s disease, FTD; Frontotemporal dementia, PD; Parkinson’s disease, LBD; Lewy body disease, HD; Huntington’s disease.

## 4. The Association between GSH Dysregulation and MSA

The fact that mutations of COQ2, which is an important factor for regulating redox states, were found in specific populations of patients with MSA [70] suggests that oxidative stress in the brain is associated with the cause or progress of the disease. The brain might be inherently more vulnerable to oxidative damage due toROS, since the brain consumes more oxygen to produce energy—and consequently generates higher levels of toxic ROS—compared to other organs [71]. In addition, the brain contains an abundance of lipids with unsaturated fatty acids that act as a source of peroxidation [72]. Both features indicate the importance of neuroprotective antioxidants in the context of MSA and other NDs. 

### 4.1. GSH Levels in Patients with MSA

GSH is one of the most effective antioxidants against oxidative stress in the brain [73]; other antioxidants, such as catalase, superoxide dismutase (SOD) and glutathione peroxidase (GPx), are present but expressed at lower levels there [74,75]. GSH detoxifies all types of ROS, including superoxide anions (O_2_^−^), hydroxyl radicals (^•^OH), hydroperoxyl radicals (HO_2_^•^) and hydrogen peroxide (H_2_O_2_), and also acts as an electron donor, resulting in disulfide bond formation to produce oxidized glutathione (GSSG) [76]. GSSG is a substrate of the flavoenzyme glutathione reductase (GR), which transfers an electron from nicotinamide adenine dinucleotide phosphate (NADPH) to GSSG, thereby regenerating GSH and constituting a system for recycling GSH [77]. A trend of decreased levels of nigral GSH has been reported in the postmortem brains of patients with MSA compared to healthy controls, although the difference did not reach statistical significance [78,79]. Another report found a lower ratio of reduced GSH to oxidized GSSG, an index of oxidative stress, in the substantia nigra of patients with MSA, although, the change was not statistically significant [80]. These reports suggest that a decline in GSH exacerbates oxidative stress in the brains of MSA patients. Interestingly, in vitro experiments showed that the amyloid formation of α-syn was significantly facilitated by GSSG while GSH suppressed aggregation [81]. Furthermore, several studies have shown that increased intracellular GSH levels partially alleviated α-syn oligomerization and its cytotoxicity [82,83,84]. These results suggest that GSH depletion followed by oxidative stress promotes the pathological fibrillation of α-syn (Figure 2).

The states of α-synuclein (α-syn) may depend on the redox states of glutathione, which are regulated by glutathione reductase (GR) and glutathione peroxidase (GPx). GR transfers an electron from nicotinamide adenine dinucleotide phosphate (NADPH) to GSSG and thereby catalyzes the reduction of GSSG to GSH. GPx reduces peroxide to a harmless compound by gathering the needed reducing equivalents from GSH. The aggregation ofα-syn is suppressed by ample GSH while the amyloid formation of α-syn is facilitated by GSSG. In addition, GPx markedly enhances the fibrillation of α-syn.

### 4.2. Possible Association of the Enzymes for GSH Synthesis and Metabolism with MSA

Synthesized GSH has been shown to react non-enzymatically with ROS or to function as an electron donor for the reduction of peroxides in the GPx reaction [85]. The formation of a disulfide bond between two GSH molecules gives rise to GSSG during this process. In a 1986 report, there were no significant alterations in GPx activity in the autopsied brains of patients with SND compared to healthy controls, suggesting that neuronal cell loss is unlikely to result from reduced activity of brain GPx [86]. However, there has been a report showing that α-syn enhanced in vitro GPx activity which in turn markedly enhanced fibrillation of α-syn [87]. Further studies at the cellular level are required to elucidate the role of GPx activity in the brains of patients with MSA.

Glutathione-S-transferases (GSTs) are a superfamily of phase 2 detoxification enzymes that detoxify both ROS and toxic xenobiotics, primarily by catalyzing GSH-dependent conjugation and redox reactions [88]. It has been reported that immunopositivity of GST in oligodendroglial inclusions did not show topographical linkage to neuronal degeneration [89]. In addition, a PubMed search revealed no studies showing dysregulation of the enzymes related to GSH synthesis and metabolism in the brains of patients with MSA. However, since the number of studies describing the physiological characteristics of MSA is still quite limited, further investigations are warranted to determine whether GSH and its related enzymes are dysregulated in the brains of patients with MSA.

### 4.3. Association between Transporters Related to GSH Biosynthesis and MSA

GSH is known to be ubiquitously distributed in many cells as a major non-protein thiol, which also functions as a storage and transport form of cysteine and an important player in antioxidative defense [90]. An excess amount of cysteine can be toxic to cells, because excess cysteine induces free radical generation and extracellular glutamate production [91,92]. Additionally, cysteine can impair mitochondrial respiration by limiting iron bioavailability through an oxidant-based mechanism [93].

GSH is a tripeptide composed of three amino acids: cysteine, glutamate, and glycine [73]. Among these amino acids, cysteine is the rate-limiting substrate for enzyme activity for GSH synthesis, because the other precursor amino acids, glutamate and glycine, have much higher intracellular concentrations than cysteine [90,94]. In agreement with these findings, it has been shown that GSH concentrations can only be elevated by increasing cytosolic cysteine availability [95]. The biosynthesis of GSH occurs via a two-step ATP-requiring enzymatic process that is catalyzed by glutamate-cysteine ligase (GCL; also known as γ-glutamylcysteine synthetase) and glutathione synthetase (GSS) [96,97]. GCL, which is composed of catalytic and modulatory subunits (GCLc and GCLm, respectively), catalyzes the formation of a dipeptide, γ-glutamylcysteine (γ-GluCys), from glutamate and cysteine, the rate-limiting step in GSH biosynthesis [96]. GSS is the critical enzyme for the second step in GSH biosynthesis, which couples γ-GluCys with glycine to generate GSH [97].

In neurons, cysteine uptake is mostly mediated through a neuronal sodium-dependent transporter known as excitatory amino acid carrier 1 (EAAC1; also known as EAAT3 or SLC1A1) [98]. EAAC1 is a member of the excitatory amino acid transporters (EAATs) that belong to the solute carrier family SLC1A. Acidic amino acid transport by the EAATs is coupled to the co-transport of three sodium ions and one proton, and the counter-transport of one potassium ion [99]. It has been reported that EAAC1 is expressed mainly in neurons and partially in subsets of oligodendrocytes, in immature oligodendrocytes, and in oligodendrocyte progenitor cells, but not expressed in mature astrocytes, in the brain [100]. EAAC1-deficient mice showed the age-dependent brain atrophy of both the hippocampal CA1 cell layer and the corpus callosum [101]. Interestingly, EAAC1 has been found to be downregulated in the postmortem brains of patients with MSA and in an animal model of MSA [18]. Alanine-serine-cysteine transporters (ASCTs) are also included among the SLC1A family members, which share sequence homology with the EAATs, and transport the neutral amino acids serine, alanine and cysteine [102]. ASCTs have been shown to participate in an amino acid antiport cycle, in which the exchange of neutral amino acids between the extracellular and intracellular sides is coupled to the co-transport of sodium ions from the extracellular side, without involvement of potassium ions or protons. Cysteine might act as a substrate of ASCT1 (also known as SLC1A4), which is mainly distributed in the astrocytes but also partially localized in neurons. Since cysteine uptake for GSH biosynthesis is mainly mediated by EAAC1, the contribution of ASCT1 to GSH generation might be limited, but could possibly support the function of EAAC1 [103] (Figure 3). Polymorphism of ASCT1 has been confirmed in some pathological conditions characterized by alterations of brain development and function [104,105,106,107,108,109]. Indeed, a study investigating oxidative-stress gene in MSA found significant associations between MSA and polymorphisms of ASCT1 [110], suggesting that the function of ASCT1 in neurons and oligodendrocytes is important for the maintenance of redox states.

## 5. The Association of miRNA Dysregulation and MSA Pathology

Increasing lines of evidence suggest that over-expression or down-regulation of non-coding RNAs is involved in the onset and progression of NDs [111]. Among several kinds of non-coding RNAs, microRNAs (miRNAs) have been the most well-studied. Accumulating evidence indicates that several miRNAs are abnormally expressed in the serum, plasma, cerebrospinal fluid, cerebellum, pons, striatum, and frontal cortex of patients with MSA [18,112,113,114,115,116,117,118,119]. 

### 5.1. Molecular Mechanisms of miRNA Biogenesis

The function of miRNAs is basically to silence target gene expressions by binding to transcripts located mainly at the 3′-untranslated regions (3′-UTR), although there are also cases in which such silencing occurs by binding to transcripts in the 5′-UTR and coding region [120]. Clustered miRNAs can either be simultaneously transcribed from single polycistronic transcripts or independently transcribed. RNA polymerase II typically transcribes the primary miRNAs (pri-miRNAs), which are cleaved by a complex called a microprocessor consisting of an RNA-binding protein known as DiGeorge syndrome critical region 8 (DGCR8) and a ribonuclease III (RNase III) enzyme called Drosha [121]. DGCR8 recognizes an N6-methyladenylated GGAC and other motifs within the pri-miRNA, while Drosha cleaves the pri-miRNA duplex at the base of the characteristic hairpin structure of pri-miRNA [122]. The small, microprocessor-generated hairpin-shaped RNAs that form the 3′ overhang, called miRNA precursors (pre-miRNAs), are exported by exportin-5 in complex with RAN-GTP and are processed by a double-stranded RNase III enzyme termed Dicer, which is complexed with the trans-activation response RNA-binding protein (TRBP) [123]. The mature miRNA duplexes are loaded onto Argonaute (AGO) family proteins in an ATP-dependent manner to form an effector complex called the RNA-induced silencing complex (RISC) [124]. One strand of miRNA is then removed from the RISC to generate the mature RISC that induces gene silencing. In the earlier versions of miRBase, the removed strand of miRNA was called the “passenger” strand and represented as miRNA*. However, since it has been reported that miRNAs from the 5′ and 3′ arms of a pre-miRNA precursor both exist, these miRNAs are now represented as miRNA-5p and miRNA-3p, respectively. Post-transcriptional regulation by the RISC complex is mediated by incomplete base-paring of miRNA-mRNA interactions, likely due to the targeting of multiple transcripts, which contributes to the complexity or redundancy of miRNA systems. Recently, multiple non-canonical miRNA biogenesis pathways have been progressively identified, and have been grouped into Drosha/DGCR8-independent and Dicer-independent pathways [125] (Figure 4).

In the canonical pathway, miRNA biogenesis begins with the transcription of primary miRNAs (pri-miRNAs), which a microprocessor composed of DGCR8 and Drosha cleaves to generate miRNA precursors (pre-miRNAs), which are hairpin forms of small RNA. Pre-miRNAs are exported by exportin-5 in complex with RAN-GTP and processed by Dicer, complexed with TRBP. In the Dicer-independent pathways, after the pre-miRNA, processed by DGCR8/Drosha, is exported, it is further processed via AGO-dependent cleavage but is not cleaved by Dicer. In the Drosha/DGCR8-independent pathway, miRNAs derived from snoRNA, tRNA, shRNA, splicing and others are exported without processing and then processed by Dicer. Through these processes, the mature miRNA is combined with members of the AGO family of proteins to form miRNA-induced silencing complexes (miRISCs).

### 5.2. Dysregulation of Genes and miRNAs in Patients with MSA

The first study of the miRNA profiles in MSA cases in comparison with controls and in transgenic (tg) models of MSA compared with non-tg mice was reported in 2014, much later than the corresponding studies in NDs such as AD, PD and ALS [18]. The screening of dysregulated miRNAs in both humans and animal models of MSA revealed that miR-96 is upregulated in these cases (probably miR-96-5p, since miR-96-3p was displayed as miR-96* in the earlier version of miRBase), resulting in downregulation of EAAC1 and a taurine transporter (TauT, also known as SLC6A6), another solute carrier family protein [18]. These proteins are particularly colocalized in neuronal and glial cells neighboring the α-syn-positive oligodendrocytes. Interestingly, we have shown that administration of an miR-96-5p inhibitor to the mouse brain improves neuroprotection against oxidative stress via upregulation of EAAC1 and the resulting increase in GSH levels [126,127]. In addition, several reports have shown that miR-96-5p expression is upregulated in the serum, striatum, and frontal cortex of MSA patients [18,128,129]. Moreover, the administration of FTY720-Mitoxy, which is known to increase the expression of neurotrophic factors in oligodendrocytes, reduced the expression of miR-96-5p in an MSA-mimicking mouse model [130]. These results suggest that agents that increase GSH levels in the brain and thereby manipulate miR-96-5p levels might be used to ameliorate the symptoms of MSA.

Another report has shown that both miR-339-5p and miR-96-5p are increased in the serum of patients with MSA [128]. In that study, miR-339-5p was shown to target strong mRNA-miRNA interactions with the RNA-binding protein neuro-oncological ventral antigen-1 (NOVA1). Interestingly, we have shown that NOVA1 regulates EAAC1 expression via glutamate transport-associated protein 3–18 (GTRAP3-18), which is an endoplasmic reticulum (ER)-localized protein and negative regulator of EAAC1, by trapping EAAC1 in the ER [127]. NOVA1 is also directly regulated by miR-96-5p, suggesting that the cysteine uptake system for GSH biosynthesis is impaired in patients with MSA.

There is another report that miR-202 (probably miR-202-3p, since miR-202-5p was displayed as miR-202* in the earlier version of miRBase) is upregulated in the cerebellums of patients with MSA, in correlation with reductions in the expression of the organic cation uptake transporter (Oct1, encoded by SLC22A1) [117]. Oct1 is known to regulate numerous target genes related to cellular stress. Deficiency in Oct1 expression results in hypersensitivity to hydrogen peroxide, which in turn elevates ROS levels. In the brain, Oct1 is expressed in the endothelial cells that form the blood–brain barrier (BBB) and may be involved in translocation of organic cations across the BBB in both directions [131]. The upregulation of miR-202-3p accompanied by a low level of Oct1 has been detected in the MSA cerebellum, which could result in decreased resistance of neurons to oxidative damage and subsequent cerebellar disease. Moreover, miR-202-3p has been reported to be linked to mitochondrial dysfunction, suggesting that increasing oxidative stress by miR-202-3p dysfunction contributes to the pathology of MSA [132].

Furthermore, miR-101 (probably miR-101-3p, since miR-101-5p was displayed as miR-101* in the earlier version of miRBase) has been observed to be upregulated in the striatum of MSA patients compared to controls [129]. Overexpression of miR-101-3p in an oligodendroglial cell line inhibited autophagy by altering the expression of autophagy-related genes that promote α-syn accumulation. In addition, we previously showed that EAAC1 expression is negatively regulated by miR-101-3p in a direct manner [126]. Interestingly, lentiviral delivery of an anti-miR-101-3p construct has been shown to reduce α-syn-induced autophagy deficits in a mouse model of MSA, indicating that anti-miR-101-3p is a promising therapeutic agent for MSA [129]. 

### 5.3. Candidate miRNA Biomarkers for MSA

Several studies have used miRNA microarrays to elucidate the profile of miRNA in patients with MSA, revealing the dysregulation of several miRNAs, some of which overlapped across the studies [133,134,135] (Table 2). The most frequently reported miRNA to be dysregulated in MSA patients (6 reports) was miR-24-3p, which has also been reported to be dysregulated in AD and PD [136,137]. It has also been shown that inhibition of the miR-24-3p “sponge”, CircRtn4—one of the CircularRNAs, decreased ROS levels concomitant with increases in SOD and GSH levels [138]. In addition, miR-24-3p was also detected as a possible regulator of GPx3 in response to oxidative stress and related pathologies [139]. Four independent miRNA microarrays revealed several dysregulated miRNAs in MSA patients: miR-19b-3p, miR-25-3p and miR-92a-3p [18,112,113,114,115,119]. miR-19b-3p has been reported as a biomarker of PD in several studies [140,141]. Interestingly, both miR-25-3p and miR-19b-3p have been detected in the serum of patients with an isolated form of REM sleep behavior disorder, representing the prodromal state of the α-synucleinopathies, suggesting that these miRNAs could be potential prognostic biomarkers for α-synucleinopathies [142]. miR-92a-3p has been reported as a dysregulated miRNA in ALS [143,144]. miR-92a-3p has repeatedly been reported in connection to oxidative stress and could be a regulator of GPx3 [139,145]. Further investigation will be needed to clarify the functions of these miRNAs and whether they might have potential as either biomarkers or therapeutic agents for MSA.

## 6. Prospective for Diagnosis and Treatment of NDs including MSA

Since no NDs, including MSA, are currently curable, radically different approaches for diagnosis and treatment are needed (Figure 5). One of the important targets for the treatment of NDs is early detection, as it may be too late to start treatment after symptoms emerge and/or α-syn deposition appears [146,147,148]. Understanding the precise mechanisms of pathogenesis and early pathological features should contribute to therapeutics of NDs in the future. One of the candidate markers for early detection of NDs is GSH levels, as decreases in specific brain areas of patients with NDs have been reported in several clinical studies, and this decline seems to occur at very early stages of the disease [79,90,149]. A recent advance in the field of proton magnetic resonance spectroscopy (^1^H-MRS) enables the measurement of in vivo GSH levels with a non-invasive technique for brain metabolite quantification [26]. According to recent clinical studies, GSH levels in the blood as well as specific areas of the brain measured by ^1^H-MRS were significantly decreased in patients with AD and mild cognitive impairment, considered early-stage AD, compared to those of healthy older-age controls [23,150,151,152]. In addition, GSH reduction rates in the frontal cortex of patients with AD were correlated with decline in cognitive functions [153]. GSH levels in the brains of ALS patients were also decreased compared to those of age-matched healthy volunteers, and the decreased GSH levels in the motor cortex and corticospinal tract were inversely correlated with disease duration or time since diagnosis [154,155]. Because the aggregation and fibrillation of α-syn seems to be promoted by GSH depletion as described in Section 4.1, the detection of α-syn forms in the brain could be a biomarker for disease progression. The new method described in Section 2.3 for diagnosing α-synucleinopathies has emerged and shows tremendous promise for advancing the clinical field. In particular, the detection of seeding properties for α-syn could be a useful tool for differential diagnosis of MSA, PD and other α-synucleinopathies, possibly for evaluating disease progress and even predicting the areas of the brain likely to be affected. Considering that the genes essential for GSH biosynthesis and α-syn formation are regulated by several miRNAs, and miRNA dysregulation is linked to NDs, the detection of abnormal miRNA expressions could also be a powerful tool for early diagnosis of NDs [20,156]. As described in Section 5.2, manipulation of miRNA expression before GSH decline and/or α-syn aggregation could provide prophylactic therapy beyond the symptom-limited therapeutic agents for NDs.

Circulating miRNAs could be useful markers for the detection of MSA at a very early stage. In addition, it may be possible to modify miRNAs before GSH decline and α-syn fibrillation, which seem to occur at early stages of disease, such that disease onset might even be blocked. It is known that miRNAs regulate the expression of genes related to α-syn formation and GSH synthesis. Recent advances in proton magnetic resonance spectroscopy (**^1^**H-MRS) technology have made it possible to detect GSH decline at an early stage of NDs. In addition, an α-syn seeding assay with biopsy samples of patients’ CSF, skin and other peripheral tissue may offer differential diagnosis for early-stage MSA.

## 7. Conclusions

Since there are far fewer cases of MSA than other NDs, and because the clinical diagnosis of MSA has proven difficult, the number of MSA studies remains small. However, accumulating evidence has gradually uncovered the genes involved and clarified the mechanism of MSA pathogenesis. As in other NDs, the pathology appears to involve a complex combination of various factors. Further investigation into the dysregulation of oxidative stress-related genes via genetic mutation or dysregulated miRNAs would provide a powerful knowledge base for the potential development of therapeutic agents for MSA. Although many dysregulated miRNAs have been identified in patients with MSA, it is still unclear whether their abnormal expression is a cause, a consequence, or a compensatory mechanism. Regardless, miRNAs could be a powerful tool as biomarkers for the differential diagnosis of MSA. Further investigations will be needed to elucidate the association between MSA and dysfunction of various miRNA regulatory mechanisms and ultimately to develop novel therapeutic agents to address these issues.

## Figures and Tables

**Figure 1 ijms-23-15076-f001:**
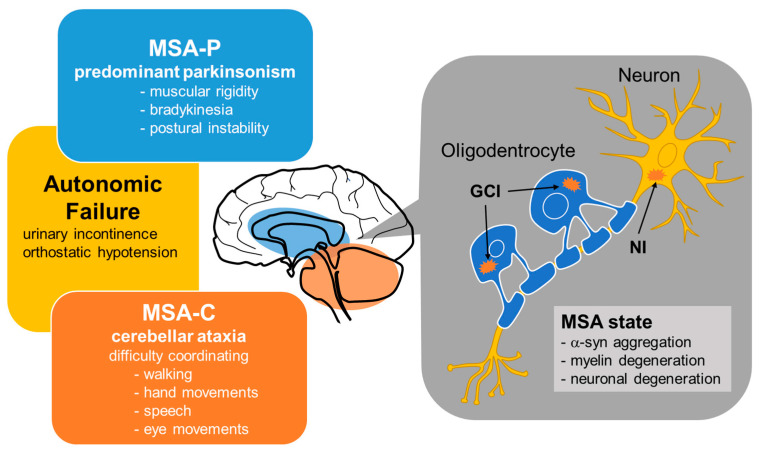
Clinical and pathological features of MSA.

**Figure 2 ijms-23-15076-f002:**
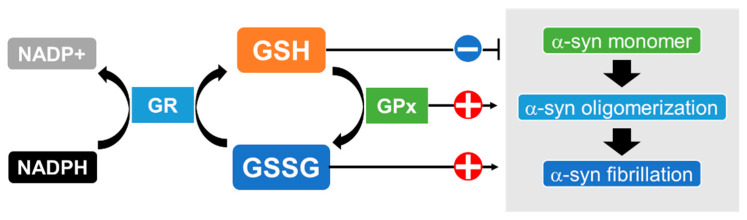
Association between the state of α-synuclein and conversion of GSH and GSSG.

**Figure 3 ijms-23-15076-f003:**
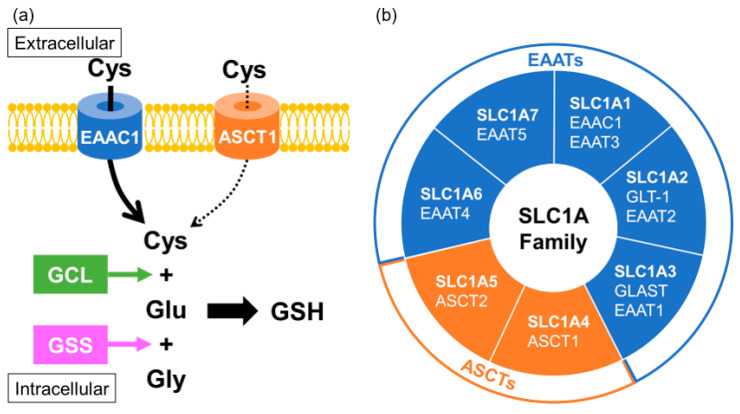
Biosynthesis of intracellular glutathione in neurons, microglia and oligodendrocytes. (**a**) Glutathione (GSH) is comprised of three amino acids: cysteine (Cys), glutamate (Glu) and glycine (Gly), of which Cys is rate-limiting, with its uptake mainly mediated by EAAC1. ASCT1 also contributes to Cys uptake, but its ability might be more limited. Transported Cys is conjugated with Glu catalyzed by glutamate-cysteine ligase (GCL) and then couples with Gly to form GSH. (**b**) EAAC1 and ASCT1 are members of the SLC1A family, comprised of excitatory amino acid transporter (EAAT) and neutral amino acid transporter (ASCT) subtypes.

**Figure 4 ijms-23-15076-f004:**
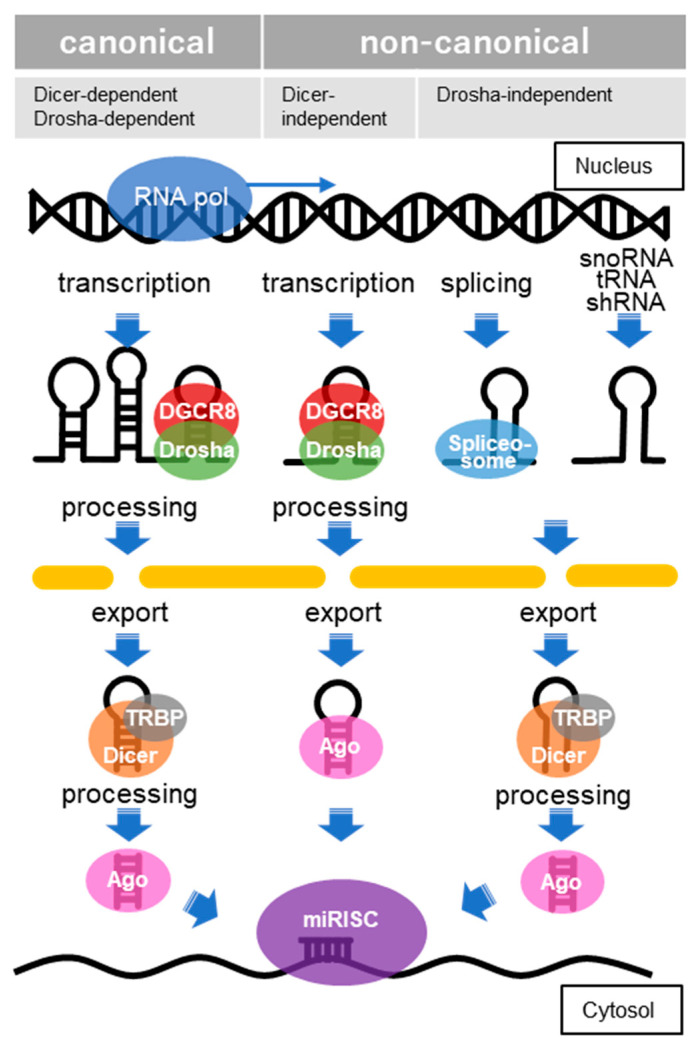
Canonical and non-canonical pathways of microRNA biogenesis.

**Figure 5 ijms-23-15076-f005:**
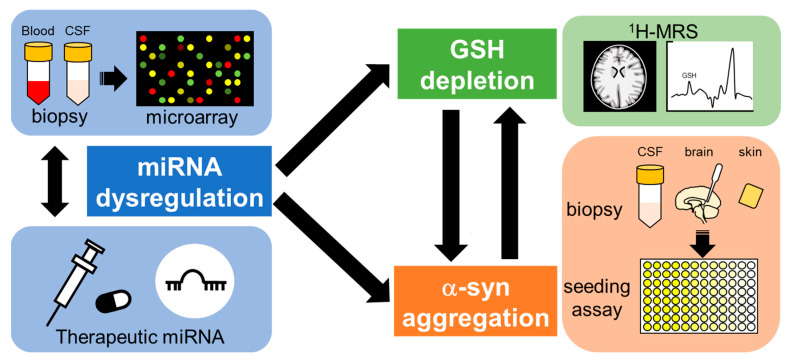
Strategies for early diagnosis and treatment of MSA.

**Table 2 ijms-23-15076-t002:** List of miRNAs that are upregulated or downregulated in patients with MSA reported in two or more studies.

miRNA	Number of Reports	Detection Area							Features	
		Serum [112,113,118]	Plasma [114,119]	CSF [119]	Cerebellum [116,117]	Pons [116]	Striatum [115]	Frontal Cortex [18]	Related NDs [133,134]	Regulatory Function [135]
hsa-miR-24-3p	6	↑↑↑	↓	n.d.	n.d.	n.d.	↑	↑	AD, PD, HD	regulation of neuronal differentiation
hsa-miR-19b-3p	4	↑	↓	↑	n.d.	n.d.	n.d.	↑	AD, PD	regulation of neural proliferation
hsa-miR-25-3p	4	↑↑	n.d.	n.d.	n.d.	n.d.	↑	↑	AD, PD	aggravating Aβ -induced neuron injury
hsa-miR-92a-3p	4	↑	n.d.	↓	n.d.	n.d.	↑	↑	AD, ALS	aggravating oxidative stress
hsa-let-7b-5p	3	↑↑	n.d.	n.d.	n.d.	n.d.	↑	n.d.	AD, PD, ALS	regulation of neuronal apoptosis
hsa-miR-15b-5p	3	↑	↓	n.d.	n.d.	n.d.	n.d.	↑	AD, ALS	protection against neuronal damage
hsa-miR-16-5p	3	↑	n.d.	n.d.	n.d.	n.d.	n.d.	↑	AD, PD, ALS	associated with Aβ deposition
hsa-miR-17-5p	3	↑↑	n.d.	n.d.	n.d.	n.d.	n.d.	↑	AD, PD	protection against neurotoxicity
hsa-miR-23a-3p	3	↑	n.d.	n.d.	n.d.	↑	n.d.	↑	ALS, HD	associated with neuronal apoptosis
hsa-miR-93-5p	3	↑	n.d.	n.d.	n.d.	n.d.	↑	↑	AD, ALS	inhibition of microglial activation and inflammatory reaction
hsa-miR-99a-5p	3	↑	n.d.	↓	n.d.	n.d.	n.d.	↑	AD, PD	localized at post-synaptic densities in forebrain
hsa-miR-106a-5p	3	↑↑	n.d.	↓	n.d.	n.d.	n.d.	n.d.	PD	regulation of neurogenesis
hsa-mIR-124-3p	3	n.d.	n.d.	n.d.	n.d.	↓	↓	↑	AD, ALS	regulation of neuronal development
hsa-miR-129-5p	3	n.d.	n.d.	n.d.	↓↓	↓	n.d.	n.d.	AD, PD, ALS	regulation of apoptosis and neuroinflammation
hsa-miR-186-5p	3	↑	n.d.	n.d.	n.d.	n.d.	↑	↑	AD, ALS	suppression of BACE1, enzyme for Aβ generation
hsa-miR-484	3	↑	n.d.	n.d.	↓↓	n.d.	n.d.	n.d.	no data	regulation of neuronal apoptosis
hsa-let-7a-5p	2	↑	n.d.	n.d.	n.d.	n.d.	n.d.	↑	AD, PD, ALS	differentiation of neural stem cells
hsa-let-7c-5p	2	↑	n.d.	n.d.	n.d.	n.d.	n.d.	↑	PD	regulation of neural stem cell differentiation
hsa-let-7d-5p	2	↑	n.d.	n.d.	n.d.	n.d.	n.d.	↑	AD, PD, ALS	regulation of neural cell fate and neurogenesis
hsa-let-7i-5p	2	↑	n.d.	n.d.	n.d.	n.d.	n.d.	↑	AD, PD, ALS	protection against brain damage
hsa-miR-20a-5p	2	↑	n.d.	n.d.	n.d.	n.d.	n.d.	↑	AD	protection against neurotoxicity
hsa-miR-21-5p	2	n.d.	n.d.	n.d.	n.d.	↑	n.d.	↑	AD, PD	protection against neuronal damage
hsa-miR-27a-3p	2	↑	n.d.	n.d.	n.d.	n.d.	n.d.	↑	AD	regulator of tight junction at brain endothelium
hsa-miR-30b-5p	2	n.d.	↑	n.d.	n.d.	↑	n.d.	n.d.	AD, PD, ALS	protection against neurotoxicity
hsa-miR-30d-5p	2	↑	n.d.	n.d.	n.d.	n.d.	n.d.	↑	AD, PD	regulation of neuronal autophagy and apoptosis
hsa-miR-96-5p	2	↑	n.d.	n.d.	n.d.	n.d.	n.d.	↑	PD, ALS	regulation of neuronal glutathione level
hsa-miR-100-5p	2	n.d.	n.d.	↓	n.d.	n.d.	↑	n.d.	AD	leading microglial accumulation
hsa-miR-103a-3p	2	↑↑	n.d.	n.d.	n.d.	n.d.	n.d.	n.d.	AD, PD, ALS	promotion neural outgrowth
hsa-miR-107	2	↑↑	n.d.	n.d.	n.d.	n.d.	n.d.	n.d.	AD	prevention of Aβ -induced neurotoxicity
hsa-miR-127-3p	2	n.d.	n.d.	n.d.	↓	↓	n.d.	n.d.	AD	regulation of neuronal autophagy
hsa-miR-129-2-3p	2	n.d.	n.d.	n.d.	↓	↓	n.d.	n.d.	no data	targeting GABA_A_ receptor to protect epilepsy
hsa-miR-130a-3p	2	↑	n.d.	n.d.	n.d.	n.d.	n.d.	↑	PD, ALS	promotion of the neuronal differentiation
hsa-miR-132-3p	2	n.d.	n.d.	n.d.	↓↓	n.d.	n.d.	n.d.	AD, PD, ALS, HD	regulation of neuronal differentiation, maturation and functioning
hsa-miR-138-5p	2	n.d.	n.d.	n.d.	↓	↓	n.d.	n.d.	AD	control of hippocampal interneuron function
hsa-miR-142-5p	2	n.d.	n.d.	n.d.	↑	n.d.	n.d.	↑	AD, PD	improvement of neural differentiation
hsa-miR-155-5p	2	n.d.	n.d.	n.d.	n.d.	n.d.	↑	↑	AD, ALS	pro-inflammatory mediator of the central nervous system
hsa-miR-181a-5p	2	↑	n.d.	n.d.	n.d.	n.d.	↑	n.d.	AD, PD	promotion of neuronal degeneration
hsa-mIR-185-5p	2	↑	n.d.	n.d.	n.d.	n.d.	↑	n.d.	AD	inhibition of neuronal autophagy and apoptosis
hsa-miR-191-5p	2	↑	n.d.	n.d.	n.d.	n.d.	n.d.	↑	AD	alleviation of microglial cell injury
hsa-miR-219a-2-3p	2	n.d.	n.d.	n.d.	↓	n.d.	↑	n.d.	no data	unknown
hsa-miR-339-5p	2	↓↓	n.d.	n.d.	n.d.	n.d.	n.d.	n.d.	AD	negative regulation of BACE1 activity
hsa-miR-371b-3p	2	n.d.	↑	n.d.	n.d.	↓	n.d.	n.d.	no data	unknown
hsa-miR-380-3p	2	↓	n.d.	n.d.	↓	n.d.	n.d.	n.d.	AD	enhancement of neurotoxicity
hsa-miR-425-5p	2	↑	n.d.	n.d.	n.d.	n.d.	n.d.	↑	AD, ALS	promotion of neuronal necroptosis
hsa-miR-486-5p	2	↑	n.d.	n.d.	↓	n.d.	n.d.	n.d.	ALS	targeting NeuroD6, scavenger gene of ROS
hsa-mIR-539-5p	2	n.d.	n.d.	n.d.	↓	n.d.	↓	n.d.	no data	inhibition of inflammatory response of neuron
hsa-miR-1233-3p	2	n.d.	n.d.	n.d.	↓	↓	n.d.	n.d.	no data	unknown
hsa-miR-1290	2	n.d.	n.d.	n.d.	↑	↑	n.d.	n.d.	no data	regulation of neuronal differentiation
hsa-miR-3663-5p	2	n.d.	n.d.	n.d.	↓	↓	n.d.	n.d.	no data	unknown
hsa-miR-4428	2	n.d.	n.d.	n.d.	↑	↑	n.d.	n.d.	no data	unknown
hsa-miR-4440	2	n.d.	n.d.	n.d.	↓	↓	n.d.	n.d.	no data	unknown
hsa-miR-4726-3p	2	n.d.	↓	n.d.	n.d.	↓	n.d.	n.d.	no data	unknown
hsa-miR-4739	2	n.d.	n.d.	n.d.	↓	↓	n.d.	n.d.	no data	unknown

Upward (red highlight) and downward arrows (blue highlight) indicate increased and decreased levels, respectively. The number of arrows represents the number of reports showing miRNA array data for serum, plasma, cerebrospinal fluid (CSF), cerebellum, pons, striatum, or frontal cortex. Abbreviations: Aβ; β-Amyloid, BACE1; β-site Amyloid precursor protein cleaving enzyme 1, GABA; γ-amino butyric acid, AD; Alzheimer’s disease, PD; Parkinson’s disease, ALS; Amyotrophic lateral sclerosis, HD; Huntington’s disease, n.d.; not detected.

## Data Availability

Not applicable.

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
