# Peer review of "Glutathione Depletion and MicroRNA Dysregulation in Multiple System Atrophy: A Review"

_ijms, 2022, doi:10.3390/ijms232315076_

Round 1
Reviewer 1 Report
This is a nice and comprehensive review summarizing the effect of miRNA dysregulation and oxidative stress in the context of MSA.
MSA and Parkinson's disease, although belong to the same class of disease, have been shown to be very different in their respective manifestation. In regards to this, the authors need to highlight an exclusive section dedicated to the discussion of different synuclein pathology in MSA versus PD, and what has been done in recent years to understand the biology of synuclein. This is particularly relevant to conformational studies showing the difference between MSA and PD aggregates, and if there are any potential links to miRNA deregulation and GSH involvement.
Can the authors provide a diagram illustrating the biogenesis and processing of miRNAs in general? It will be beneficial for the general audience to understand biology.
A table summarizing the different genes and their relationship to MSA would be helpful. It will be good to include some of the lesser-known genes that have been implicated in MSA also, like SLC1A4, NMD3 to name a few.
In the table for the miRNAs, the authors can list the function of each miRNA for the ones that have been reported, to understand the relationship of the miRNAs to the pathogenic mechanism. (Suggestion)
One of the biggest challenges in the field is distinguishing PD and MSA patients at an early stage to aid treatment and therapeutic development. It would be great to include a separate section where the authors can discuss the importance of the genetic analysis and also miRNA levels as a potential tool for the development of biomarkers (and therapeutic approach) specific to MSA and not PD. Discuss the translatability of these areas in the context of MSA (including the drawbacks)
It would also be useful if the authors can discuss and delineate any specific genes that have been implicated in MSA only and not PD.
Overall, the review is well-written and very comprehensive. The table on the miRNA levels is very comprehensive and provides a great summary of what has been done.
Author Response
We are grateful to Reviewer#1 for the insightful comments and useful suggestions. As indicated in the responses that follow, we have taken all these comments and suggestions into account in the revised version of our paper.
Comments 1: MSA and Parkinson's disease, although belong to the same class of disease, have been shown to be very different in their respective manifestation. In regards to this, the authors need to highlight an exclusive section dedicated to the discussion of different synuclein pathology in MSA versus PD, and what has been done in recent years to understand the biology of synuclein. This is particularly relevant to conformational studies showing the difference between MSA and PD aggregates, and if there are any potential links to miRNA deregulation and GSH involvement.
Response: In accord with this suggestion, we added new section describing the distinction of MSA and PD, especially focused on a-synucleopathies (section 2.3, line 136 to 158).
Comment 2: Can the authors provide a diagram illustrating the biogenesis and processing of miRNAs in general? It will be beneficial for the general audience to understand biology.
Response: In accord with this suggestion, we added figure 4, illustrating the basic mechanism of miRNA biogenesis in canonical and non-canonical pathway.
Comment 3: A table summarizing the different genes and their relationship to MSA would be helpful. It will be good to include some of the lesser-known genes that have been implicated in MSA also, like SLC1A4, NMD3 to name a few.
Response: In accord with this suggestion, we added table 1, summarizing the genes related to MSA.
Comment 4: In the table for the miRNAs, the authors can list the function of each miRNA for the ones that have been reported, to understand the relationship of the miRNAs to the pathogenic mechanism. (Suggestion)
Response: In accord with this suggestion, we added brief description of dysregulated miRNAs in MSA in table 2.
Comment 5: One of the biggest challenges in the field is distinguishing PD and MSA patients at an early stage to aid treatment and therapeutic development. It would be great to include a separate section where the authors can discuss the importance of the genetic analysis and also miRNA levels as a potential tool for the development of biomarkers (and therapeutic approach) specific to MSA and not PD. Discuss the translatability of these areas in the context of MSA (including the drawbacks)
Response: In accord with this suggestion, we added new section describing new approaches for MSA therapeutics (section 6, line 453 to 484). We also added figure 5 illustrating new methods for MSA diagnosis and treatment.
Comment 6: It would also be useful if the authors can discuss and delineate any specific genes that have been implicated in MSA only and not PD.
Response: Although we have discussed about dysregulation and dysfunction of specific genes and miRNAs in MSA in the previous version of manuscript, we separated the section 5.2 to describe expected contribution of specific genes to MSA pathology (section 5.2, line 373 to 423) according to your comment.
Reviewer 2 Report
The present review seems potentially intriguing and suggestive, this reviewer also felt concerns of their way of the presentation and explanation in the manuscript.
1) The authors did not mention about the possible explanation of selective involvement of brain regions in this disorder. This should be discussed in someway in the manuscript.
2) It was quite difficult to follow authors explanation of ASC1 and MSA developement. It would be better to show comprehensive figure illustration.
3) As Compagnoni GM reported, authors should discuss about the possible main production sites for oxidative stress and make comment of the role of mitochondria in this disorder.
Author Response
We are grateful to Reviewer#2 for the thoughtful comments. As indicated in the responses that follow, we have taken all these comments and suggestions into account in the revised version of our paper.
Comment 1: The authors did not mention about the possible explanation of selective involvement of brain regions in this disorder. This should be discussed in someway in the manuscript.
Response: In accord with this suggestion, we added latest finding that indicate distinct strains of a-syn may be associated with disease propagation and selective atrophy region as described in section 2.3 (line 136 to 158).
Comment 2: It was quite difficult to follow authors explanation of ASC1 and MSA developement. It would be better to show comprehensive figure illustration.
Response: In accord with this suggestion, we added figure 3 illustrating the mechanism of EAAC1 and ASCT1 in cysteine uptake and GSH biosynthesis.
Comment 3: As Compagnoni GM reported, authors should discuss about the possible main production sites for oxidative stress and make comment of the role of mitochondria in this disorder.
Response: In accord with this suggestion, we added the description of oxidative stress and major source of ROS in section 2.2 (line 111 to 126)
Round 2
Reviewer 2 Report
Authors successfully manage all the criticisms raised by this reviewer.